# Do Teachers Question the Reality of Pain in Their Students? A Survey Using the Concept of Pain Inventory-Proxy (COPI-Proxy)

**DOI:** 10.3390/children10020370

**Published:** 2023-02-13

**Authors:** Rebecca Fechner, Melanie Noel, Arianne Verhagen, Erin Turbitt, Joshua W. Pate

**Affiliations:** 1Graduate School of Health, University of Technology Sydney, Sydney, NSW 2007, Australia; 2Department of Psychology, The University of Calgary, 2500 University Drive NW, Calgary, AB T2N 1N4, Canada; 3Alberta Children’s Hospital Research Institute, 2888 Shaganappi Trail, Calgary, AB T3B 6A8, Canada; 4Hotchkiss Brain Institute, 3330-3330 Hospital Dr Nw, Calgary, AB T2N4N1, Canada

**Keywords:** pain science education, teachers, pediatrics, adolescents, chronic pain, stigma, concept of pain

## Abstract

An assessment of a teacher’s concept of their student’s pain could be useful to guide preventative and targeted school-based pain science education. We aimed to assess a teacher’s own concept of pain against their concept of their student’s pain and examine the psychometric properties of the tool. Teachers of 10–12-year-old children were invited to participate in an online survey via social media. We modified the Concept of Pain Inventory (COPI) by inserting a vignette (COPI-Proxy), and we included questions to explore teacher stigma. Overall, a sample of 233 teachers participated in the survey. The COPI-Proxy scores showed that teachers can conceptualize their student’s pain separately but are influenced by their own beliefs. Only 76% affirmed the pain in the vignette as real. Teachers used potentially stigmatizing language to describe pain in their survey responses. The COPI-Proxy had acceptable internal consistency (Cronbach’s alpha = 0.72) and moderate convergent validity with the COPI (r = 0.56). The results show the potential benefit of the COPI-Proxy for assessing someone’s concept of another’s pain, particularly for teachers who are important social influencers of children.

## 1. Introduction

School attendance positively affects health and wellbeing in children [1]. However, children experiencing chronic pain, defined as pain lasting longer than 3 months, have an average absence from school of 22% [2]. Teachers and school staff are well positioned to influence how chronic pain is conceptualized and managed, given that students spend 40% of their awake time at school per week. Despite this, there is limited evidence to identify effective responses from schools and teachers when managing chronic pain in their students [3] and supporting inclusion for affected students [2]. 

Teachers may hold biases about chronic pain that could extend to their perceptions of and attitudes toward their students [4]. These biases and the resulting attitudes and responsiveness to pain may affect ongoing trajectories for how children and adolescents perceive pain as they age [5]. Pain-related stigma has been reported by students suffering with chronic pain in the school environment, with identified sources of stigma being pain invisibility, lack of chronic pain knowledge, and lack of understanding from teachers and peers [6]. Reconceptualizing one’s concept of pain using pain science education (PSE) shows promise in the literature for adults [7,8,9] and children [10,11,12]. However, it is currently unknown whether a teacher’s own concept of pain affects their concept of their student’s pain and whether reconceptualization is necessary for them to respond in a way that positively influences stigma.

The Neurophysiology of Pain Questionnaire (NPQ) has been used to assess parental pain knowledge in one study which involved the delivery of pain science education to children [13]. Originally the researchers of that study planned the development of a pediatric NPQ (PedNPQ). However, the PedNPQ had poor test–retest reliability and only slight-moderate inter-rater reliability. Hence, the research team reported it as not valid [13]. 

The Concept of Pain Inventory (COPI) is a recently developed tool used to assess a child’s concept of pain, which allows clinicians and researchers to individualize education and target pain beliefs [14]. The COPI was recently recommended as a superior alternative to the NPQ in future studies involving PSE in pediatrics [13], because it was developed for children using simpler language and a five-point Likert scale to assess the level of agreement, rather than True/False, which indicates correctness. The COPI has been validated in children aged 8–12 years, with acceptable internal consistency (Cronbach’s α = 0.78) and moderate test–retest reliability (intraclass correlation coefficient (3,1) = 0.55; 95% CI, 0.37–0.68) [14]. We deemed it an appropriate tool to modify as a proxy to assess one’s concept of another person’s pain, such as a teacher of their student’s, or a parent of their child’s, because it allows for a direct item-level comparison between children and their adult influencers when treating these dyads. We called the modified questionnaire the COPI-Proxy (Appendix A).

Given a teacher’s role as an educational expert who responds to and scaffolds pain-related developmental beliefs and skills for children in their everyday interactions [5,15,16], we were interested in assessing and understanding a teacher’s concept of their student’s pain. Thus, we hope to better inform potential educational interventions involving adult influencers of children, such as pain science education for teachers in schools. A necessary prerequisite to guide these interventions is to first assess the usability and validity of the COPI-Proxy in this population. In this study the aims were to: (1) examine the distribution of total and item-level scores for the COPI and COPI-Proxy and compare these distributions in a population of teachers that teach 10–12-year-old children, and (2) evaluate the psychometric properties of the COPI-Proxy by assessing the factor structure, internal consistency, and convergent validity. 

## 2. Materials and Methods

This online survey study was approved by the University of Technology Sydney Human Research Ethics Committee (REF: ETH21-6599). We also applied to 16 Australian school ethics jurisdictions to allow us to approach school principals to distribute the survey, and approval was obtained from Melbourne Archdiocese Catholic Schools (MACS 1191-NAF), Brisbane Catholic Education (AppID: 510), the Queensland Department of Education (550/27/2586), the South Australian Department of Education (2022-0011), and the Western Australia Department of Education (170522). We preregistered the protocol before the study commenced at https://osf.io/vknj7 (accessed on 21 October 2021). We used the COSMIN checklist to guide study design and reporting [17].

### 2.1. Participants and Procedures

We invited teachers to participate in this survey using a link embedded in a video-based social media post on Twitter, Instagram, and Facebook. Recruitment using snowballing [18] was attempted by sharing the flyer via the Pain in Childhood email list https://listserv.dal.ca/index.cgi?A0=PEDIATRIC-PAIN (accessed on 12 December 2021) and requesting that these colleagues share the flyer with their friends and colleagues to promote increased visibility. We also shared the flyer via email to school principals for distribution with their staff in jurisdictions we had ethical approval to approach. School teachers were eligible if: (1) they held current registration to work as a teacher in Australia, New Zealand, the United Kingdom, Canada, or the United States of America, and (2) they were currently teaching children aged 10 to 12 years. We specifically chose these countries due to similarities in teacher training, work conditions, and language. The age range of 10–12 years was chosen because we wanted to target teachers that would likely spend more time with students each day (i.e., students still in primary school with a single teacher), and this is within the age range of 8–12 years for which the COPI is validated [14]. We chose the upper age range for those validated for the COPI because chronic pain is more prevalent with increasing age [19]. We restricted the age range to 2 years because children vary in how they communicate pain depending on their developmental stage, and we wanted to increase the relevancy for a vignette of a 12-year-old student. The first page of the survey was an informed consent page.

### 2.2. Data Collection

There were 3 parts to the survey: (1) general participant questions; (2) the COPI; and (3) the COPI-Proxy which included a vignette. We pilot tested the survey by inviting five teachers who fit the inclusion criteria to review the initial draft wording. We asked them to provide feedback on (1) relevance, (2) face validity, and (3) usability of the entire survey. Responses were pooled and presented to the research team to reach consensus on the final wording of the survey and vignette.

#### 2.2.1. General Participant Questions

The survey consisted of demographic and exploratory questions designed to understand the pool of participants. Teachers are a unique population with a recognized set of skills and knowledge related to understanding child development and responding to their students. Because of this, we hypothesized that their COPI-Proxy scores might be influenced by unique psychosocial variables, and included questions in our survey to explore this.

Demographic questions included gender, teaching experience, and the presence of their own chronic pain and location. Exploratory questions related to their specific beliefs and teaching strategies related to chronic pain (e.g., ‘What teaching and behavioral strategies have you used to support children with chronic pain in the classroom?’). Exploratory questions were a mix of yes/no response, choosing from a list, and open response.

#### 2.2.2. The Concept of Pain Inventory: COPI

The (COPI) is a 14-item questionnaire which has been developed using expert and patient input and validated in children aged 8–12 years [14]. Each item is rated on a 5-point (0–4) Likert scale ranging from 0 = ‘Strongly disagree’ to 4 = ‘Strongly agree’, with 2 = ‘Unsure’. A total score is calculated (0–56), where higher scores reflect knowledge and beliefs more closely aligned with contemporary pain science.

Following the COPI questions, a clarifying question was asked about whether they responded to the COPI according to their own experiences. This question was included to account for whether the questionnaire had been completed as intended (that is, in relation to their own pain). 

#### 2.2.3. The Concept of Pain Inventory-Proxy: COPI-Proxy

In this part of the survey participants were directed to a 20 s video vignette. The video presents a scenario with a child named Sarah experiencing recurrent stomach pain. She presents on that day with stomach pain after lunch and is crying and requesting to go home. The COPI-Proxy questionnaire includes all COPI items and is modified to read like a proxy version for a teacher (or another person) to complete in relation to the child in the vignette (Sarah). Apart from inserting the child’s name, Sarah, into each item, minor grammatical edits were made so that each item is clearly in relation to Sarah’s pain. Items are rated and scored as per the COPI.

### 2.3. Sample Size

A target sample size of 200 and a minimum sample size of 100 respondents was determined based on the COSMIN guide for factor analyses (seven times the number of items and ≥100) [17].

### 2.4. Data Analysis

Survey data were collected and managed using REDCap electronic data capture tools [20,21] hosted at University of Technology Sydney. Participant enrollment occurred from December 2021 to August 2022. 

To describe our participant sample, we used descriptive statistics. For list-based questions we used content analysis [22]. For short answer questions we used thematic analysis with a codebook approach [23] and summarized these findings in a table. 

To describe the scores on the questionnaires, we first used the Shapiro–Wilk test to investigate the normality of the distribution of COPI and COPI-Proxy scores. When scores were not normally distributed (Shapiro–Wilk’s tests *p* < 0.05) we presented medians and interquartile ranges (IQR). Items where >90% of respondents “strongly agreed” or “strongly disagreed” were identified as having possible ceiling or floor effects. Distributions for COPI and COPI-Proxy totals and item-level scores were represented on stacked bar graphs, to visually convey the responses.

For the psychometric properties of the COPI-Proxy, we first assessed the factor structure. A maximum likelihood factor analysis with oblimin rotation was conducted to extract potential factors. Oblimin rotation was used to allow for correlation. A scree plot was used to indicate the number of factors that should be generated by the analysis. Factor solutions were forced based on visual inspection of a scree plot (i.e., the elbow on the scree plot indicates the number of factors), theoretical groupings, and how well items loaded onto resulting factors [24]. 

To assess internal consistency, Cronbach’s alpha was calculated with values of <0.7 indicating low internal consistency, ≥0.7 and ≤0.9 acceptable internal consistency, and >0.9 potential redundancy or duplication [25]. 

To assess convergent validity between the COPI and COPI-Proxy, we calculated a Pearson’s correlation [26]. Correlation coefficients were interpreted as follows: none to low r = 0.3, moderate 0.3–0.7, and a high correlation r > 0.7 [27]. Because teachers may relate their concept of pain to that of their student, but also separate their own beliefs from their teaching strategies, we hypothesized a priori that the correlation between the total scores would be moderate. We also calculated an item-level analysis to identify variation between the COPI and COPI-Proxy item scores. At an item level, if the COPI and COPI-Proxy responses varied by 2 points or more, we deemed this an important difference. To explore possible relationships between COPI-Proxy scores and other generic variables (teacher gender, years of experience, chronic pain status, abdominal pain (because the student in the vignette had abdominal pain), COPI scores, encountering pain in the classroom, and thinking that students in pain should still attend school), we performed unadjusted univariate regression analyses. 

All data were analyzed using SPSS v25.0.0.1 [28].

## 3. Results

### 3.1. Participants

Accurate data on how many people viewed the social media post could not be collected. For example, the tweet was viewed 501 times at the end of the data collection period, but this may not indicate how many people read the whole invitation. Of the 393 teachers clicking on the survey link, 326 met the eligibility criteria, and of these, 324 (99%) consented and 233 (72%) completed the questionnaire. 

#### 3.1.1. General Participant Questions

Participants were predominately females with more than 15 years of teaching experience (Table 1). Approximately half of participants (44%) reported their own chronic pain, and the most common sites were the back and shoulder/neck. In this sample of teachers, 79% had encountered students with recurrent or chronic pain in their classroom before and 76% thought Sarah’s pain was real. Just over half of our sample (58%) thought that pain should not necessarily stop a student from attending school.

#### 3.1.2. Participant Exploratory Questions

The results of exploratory questions that we asked teachers are presented in Table 2, Table 3 and Table 4. Table 2 summarizes the responses to the question: ‘What do you think could be the cause of Sarah’s pain?’. Initially participants provided open-ended responses, and results from the content analysis are reported as percentages. Of the 181 participants that completed the question, 75% reported anxiety as one possible cause. Food allergies, menstruation, social issues (e.g., friendship troubles, family conflict), and stomach illness were also common responses. Only 1% of teachers responded that Sarah might be avoiding classwork. Participants were then asked the same question but given a list of possible causes to choose from. Anxiety as one possible cause was checked by 92% of teachers, 65% thought she might not like the afternoon classes, and 20% responded that she might be faking her pain.

A summary of thematic analysis and sample quotations from participants when answering the question ‘Please tell us more about why you think Sarah’s pain is real or not real?’ is represented in Table 3. Two main themes were identified: (1) The meaning and purpose of Sarah’s pain as criteria for ‘real’ (i.e., teachers might consider the presence of physical symptoms in the body when trying to decide if pain is real) and (2) The observed pattern and associated behaviors of the student’s pain that teachers observed (e.g., When did it happen? Was the child crying?). Teachers that answered ‘Yes’ to the question explained their reasoning differently to those who answered ‘No’ or ‘It depends’, but the underlying themes were consistent.

Regardless of affirming the pain as real, teachers listed many teaching and behavioral strategies for managing pain in the classroom. These strategies influence a student’s social and emotional wellbeing, their environment or physical comfort, and their sensory and cognitive loads. Teachers typically answered with multiple strategies (all but ten listed more than one strategy). Aside from sending the student to the school nurse (or sick bay), examples include taking short breaks, adjusting the position/environment or work expectations, partnering with peers for difficult tasks, scaffolding play conversations, and communicating with parents.

When considering teacher responses to whether students with pain should not attend class, again two main themes were identified: (1) The teacher’s role and its boundaries (i.e., their understanding of their role in attending to a student’s wellbeing) and (2) How pain influences learning and the wider classroom (e.g., Does a student have to be pain-free to learn? What about the other students?) (Table 4).

### 3.2. Distribution of COPI and COPI-Proxy Scores

The Shapiro–Wilk test was *p* < 0.05, so results are presented as median and IQR. The median COPI-Proxy total score was 40 (IQR 37–43; max 56). The median COPI total was also 40 (IQR 37–43; max 56) (Table 1). All items generally had high levels of agreement. No items had a ceiling/floor effect of >90% ‘Strongly agree’ or ‘Strongly disagree’. The ratings of agreement for each COPI and COPI-Proxy item are shown in Figure 1. The COPI-Proxy item which most participants disagreed/strongly disagreed (12%) with or were unsure (31%) about was ‘Learning about pain would help Sarah feel less pain’. Overall, teachers had more agreement with COPI-Proxy items relating to Sarah’s pain than with COPI items.

### 3.3. Psychometric Properties

#### 3.3.1. Factor Structure of the COPI-Proxy

The visual inspection with scree plot elbow criteria (i.e., the point at which the downward curve levels off) suggested one factor similar to that of the COPI; however, three items in the COPI-Proxy did not load well onto this single factor (factor loading <0.32) [29]: Item 5: “Sarah’s pain is a warning that her body needs to be protected” (factor loading 0.14); Item 9: “Sarah can have an injury and feel no pain” (factor loading 0.18); and item 13: “Pain usually feels better if Sarah moves her body a little bit more each day” (factor loading 0.28).

#### 3.3.2. Internal Consistency of the COPI-Proxy

The internal consistency of the COPI-Proxy in this sample was acceptable (Cronbach’s alpha = 0.72).

#### 3.3.3. Convergent Validity between COPI-Proxy and COPI Scores

We found moderate correlation (r = 0.56) between COPI-Proxy and COPI scores. At an item level, only three teachers did not vary at all in their responses between the COPI and COPI-Proxy items. Figure 2 illustrates the item scores and variation between teacher responses, the direction of variation, and magnitude of difference. A total of 133 participants (57%) varied by 2 points or more between the COPI and COPI-Proxy in at least 1 item. Of these, 58 (25%) differed by ≥2 points in 1 item, 40 (17%) in 2 items, and 34 (15%) in 3 or more of the 14 items.

The relationship between variables explored in the survey and COPI-Proxy scores are represented in Table 5. For every point higher a teacher scored on the COPI, they scored 0.5 points higher on the COPI-Proxy. Teachers who thought that Sarah’s pain was real scored 1.4 points higher on the COPI-Proxy. Teachers who answered ‘no’ to whether students with chronic pain should not attend class (i.e., ‘No, they should stay in class’) scored 1.2 points higher on the COPI-Proxy. Gender, teaching experience, chronic pain status, abdominal pain, and whether teachers had encountered chronic pain in their class before did not have a significant relationship to higher COPI-Proxy scores.

## 4. Discussion

Our results indicate that it is possible and indeed common for teachers to hold separate beliefs about their own pain to that of their student’s pain, with some variation in responses in all but three teachers. Variation of item-level scores by ≥2 points occurred in more than half of the teachers in one or more corresponding COPI and COPI-Proxy items. An exploratory factor analysis suggests that the COPI-Proxy measures a single construct: a teacher’s concept of their student’s pain. The COPI-Proxy has acceptable internal consistency, and moderate convergent validity with the COPI, highlighting that there is potential benefit to using the COPI-Proxy to assess the concept of another’s pain for teachers, who are important influencers to their students.

Previous research indicates the presence of pain-related stigma in schools, where teachers’ perceptions of and responses to children with pain relate to whether there is medical evidence for a pain condition [6,30,31,32,33,34]. Our vignette did not have a medical explanation to the pain. Like other studies, teachers in our study had widely varying criteria for how pain is perceived as ‘real’ or not [31]. One teacher responded, “It could be real and be a stress response to the situation that is causing her anxiety, or it could be that she is faking feeling sick to get out of the social situation that is making her feel anxious”. This juxtaposition between a potential understanding that many things can influence a pain experience, and the endorsement of language that siloes psychosocial factors as a cause for pain that is ‘not real’ or ‘fake’, could be important to consider when addressing how pain is portrayed and communicated in classrooms. Teachers may be negatively influencing pain-related trajectories and outcomes for students. Inconsistency in language may reflect a possible gap in understanding the complexity of pain, which could be contributing to pain-related felt stigma in students who experience chronic pain [4,6,30,32], despite the best intentions of teachers to scaffold and support their students. Pain science education may provide teachers with a deeper understanding and consistent language to communicate pain experiences, which in turn may reduce stigma in schools.

School attendance plays an important role in treating chronic pain through exposure to developmentally appropriate cognitive, social, physical, and emotional opportunities [3]. Teachers in our study reported using adaptive and inclusive strategies to support attendance [33], reflecting findings from other studies where teachers appear to have a biopsychosocial understanding of pain [3,34,35]. However, despite an abundance of strategies at hand, teachers may not employ these strategies consistently. For example, responses from teachers such as: “I am a teacher not a Dr” and “As teachers one of our roles is to support our students to understand what causes them stress or worry” illustrates the variation in how teachers view their role in pain interactions with students. These findings are comparable to other studies exploring teacher roles in responding to students with pain [15,32,34,35]. The results of our research indicate that perhaps there is a need for teachers to have a deeper understanding of pain to feel autonomous in their role.

The median (and IQR) of the COPI in teachers was very similar to parent results in another study that used the same COPI version [36]. These results may indicate that despite the requirement of teachers to respond to, teach, and model health and wellbeing concepts as part of their day-to-day teaching, they may not be any more equipped than the parents of the children they work with in terms of how well their concept of pain aligns with contemporary pain science [14,36,37]. Given that more than half the teachers had clinically relevant item-level variation between their COPI and COPI-Proxy responses and that there were no items with ceiling effects, there is scope for pain science education to have potential benefits for teachers. In recognition of their social influence, this aligns with recommendations for parents for the treatment of pediatric chronic pain [38].

### 4.1. Strengths and Limitations

There are several strengths and limitations of this study. The large sample size of teachers allowed for appropriate statistical analyses of psychometric properties [17]. However, due to the nature of online surveys, there is potential for bias due to self-selection, which means we cannot assume that this population is representative of the larger population of teachers. Whilst we planned our survey to be consistent across teachers with similar training and teaching environments, this provided some limitations in that the sample only represented teachers in these geographical locations. Repeating the study in other languages and other geographical locations would better represent teachers around the world. We also carefully planned our sample to address the specific age range of 10–12 years. Repeating the study in other age categories would increase overall understanding of the teacher/student relationship at different stages of development.

Parts of our survey were exploratory, with the aim of understanding our findings. Therefore, our analyses of open responses should be interpreted with some caution, as teachers were only given the opportunity to report short answers, and answering these questions was not mandatory. Further qualitative research to explore lived experience with teachers and students in the classroom would be beneficial to support these results.

Factor analysis revealed that 11 of the 14 items loaded well onto the 1 factor solution; however, we did not conduct an item-level reduction process because we wanted to compare all corresponding items of the COPI-Proxy with the COPI. Future studies could explore item reduction and assess psychometric properties with the reduced items.

### 4.2. Clinical, Educational and Policy Implications

Using both the COPI and COPI-Proxy to assess concepts of pain in dyads involving children and their important adult influencers (such as parents, coaches, and teachers) may support health professionals to separate pain beliefs and negotiate treatment approaches in these dyads [14,36,38]. One study that assessed both parent and child COPI scores suggested that tackling differences, or ‘sticking points’, between parent and child responses at an item level could be a potential starting point for therapy [36]. The use of the COPI-Proxy may be beneficial to guide educational and therapeutic interventions regarding not only sticking points, but also the social and relational nature of experiencing pain [39]. For example, in a clinical setting, the COPI-Proxy could be tested with parents. This may enable a more individualized family-centered approach to targeting pain science education and linking to therapy focused on parent/child relationships and pain, such as how a parent conceptualizes their child’s pain, how this relates to their response to their child, and the resulting experience of their child. Relevant outcomes for this example include improved parent/child relationships, improved parental confidence to support ongoing activity participation for their child with pain, and improved quality of life.

The finding that we can potentially hold different beliefs about pain in oneself and another is important. Educational approaches may not necessarily require a complete reconceptualization of a person’s pain to change their beliefs, and hence responses, to others with pain. This could have implications for public health interventions targeted to changing attitudes to pain, such as pain science education in schools or sporting groups. The focus here could be on the social responsibility of pain within a biopsychosocial framework, rather than education focused on changing one’s own beliefs. Changing attitudes toward pain in others could improve quality of life for children experiencing, or at risk of experiencing, chronic pain at school.

A recent systematic review indicated the importance of written policies to guide teachers on how to facilitate optimal pain management in schools [3]. However, social and cultural factors, ranging from government and educational policy to economic capacity and cultural beliefs, may create a need to individualize how pain is managed within a school. The results of our study suggest that teachers use a wide variety of skills and strategies to manage pain in their classrooms, though it is unknown if they are guided by consistent policy. Future research could involve teachers in decision-making and the co-design of guiding principles and/or policy, both at a school and national level. Upskilling teachers with a contemporary understanding of pain could be a necessary first step to capitalize on their existing strengths, so that they may play a role in positively influencing school and community attitudes toward pain.

### 4.3. Research Implications

We have identified several important areas for future research. First, research could test and validate the COPI-Proxy in different populations such as parents, in languages other than English, in other countries, and in populations of teachers that teach different aged students (e.g., adolescents). Second, qualitative research could explore the lived experiences of teacher-student interactions regarding chronic pain, and, in particular, teacher-specific language, perceptions, and strengths. Third, given teachers’ potential to influence student pain experiences [40], it is possible that teachers may benefit from pain science education to support their confidence and capacity to respond consistently to pain in students, using biopsychosocially-sensitive and validating language. This needs to be evaluated in randomized clinical trials to evaluate whether large scale preventative pain science education in schools and teacher training could break the cycle of pain-related miscommunication and stigma for our future generations. Future research could explore feasibility, dose, and specificity of pain science education in schools and teacher training programs. Integrating teachers with a co-design approach to research seems key.

## 5. Conclusions

The COPI-Proxy shows promise as a useful tool to assess a teacher’s concept of their student’s pain with acceptable internal consistency and moderate convergent validity. Because social and relational factors are integral to a pain experience, the COPI-Proxy could be useful to direct treatment approaches in dyads such as teacher and student as it allows for direct item-level comparisons with the child’s concept of pain. The COPI-Proxy could also be useful to assess and direct population-level interventions such as pain science education in schools.

## Figures and Tables

**Figure 1 children-10-00370-f001:**
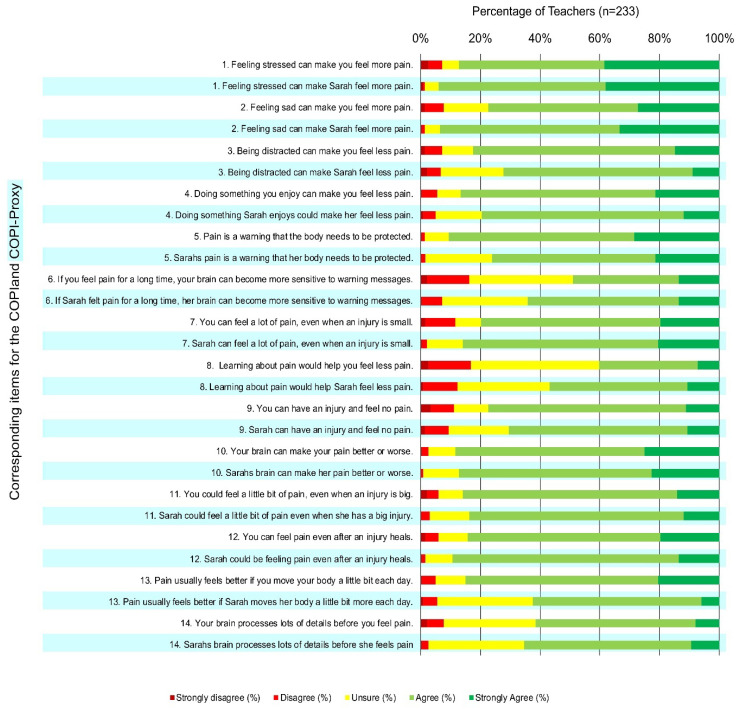
Distribution of scores for COPI and COPI-Proxy items.

**Figure 2 children-10-00370-f002:**
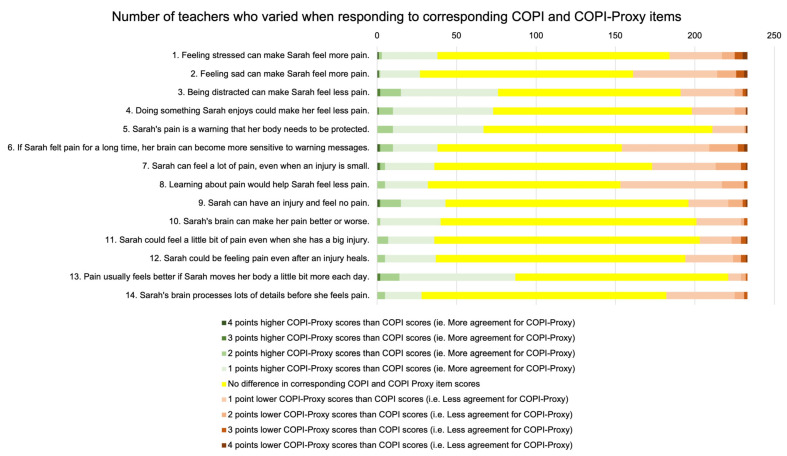
Number of teachers who varied when responding to corresponding COPI and COPI-Proxy items.

**Table 1 children-10-00370-t001:** Demographic and clinical variables of participants.

Categorical Variables	n (% of 233 Total)
Gender	
Female	215 (92)
Years teaching	
1–2 years	20 (9)
3–5 years	32 (14)
6–10 years	47 (20)
11–15 years	36 (15)
>15 years	98 (42)
Do you have persistent pain?	
Yes	104 (44)
Where have you experienced persistent pain?	
back	54 (23)
shoulder/neck	47 (20)
leg/foot	27 (12)
head	22 (9)
arm/hand	16 (7)
face/jaw	14 (6)
abdomen	13 (6)
chest	1 (0.4)
other	11 (5)
When answering questions about pain, did you reflect on your own pain experience?	
Yes	225 (97)
Do you think Sarah’s pain is real?	
Yes	176 (76)
No	2 (1)
It depends	55 (24)
Have you encountered children in your classroom that report recurrent or persistent pain?	
Yes	184 (79)
No	46 (20)
No response (optional question)	3 (1)
Do you think students that report pain should not attend class?	
Yes	32 (14)
No	135 (58)
I don’t know	66 (28)
**Continuous variables**	**Median (IQR)**
Total COPI score (0–56 *)	40 (37–43) (mean = 40.1)
Total COPI-Proxy score (0–56 *)	40 (37–43) (mean = 40.4)

Note: Characteristics may not sum to exactly 100% due to the effect of rounding. Bold indicates column headings. IQR = Interquartile Range. * Higher scores indicate greater alignment with contemporary pain science.

**Table 2 children-10-00370-t002:** Summary of responses about the possible cause of Sarah’s pain.

Question	Response	%
What do you think could be the cause of Sarah’s pain?	Anxiety	75
	Food allergy	39
	Menstruation	38
(Open response)	Social issue (e.g., friendship troubles, family conflict)	36
	Stomach illness	31
	Learning difficulty	12
	Injury (e.g., muscle strain)	11
	Trauma and/or abuse	6
	More serious medical issue	6
	Emotional/sad/upset	6
	Hungry	3
	Tired/fatigue	2
	Dehydrated	2
	Eating disorder	1
	Avoidance of classwork	1
	Pregnancy	0.5
	Insect bite	0.5
	Sexually transmitted disease	0.5
Tick all that you think might apply (for the possible cause of Sarah’s pain).	She is anxious	92
	She is being bullied	66
	She doesn’t like the afternoon classes	65
	Something she ate	59
	She has IBS	56
	She has something seriously wrong and needs to be checked by a Doctor	51
	She wants attention	39
	She is too hot after running around at lunch	32
	She is faking her pain	20
	I don’t know	13
	Other (period pain, covid, gastro)	10

**Table 3 children-10-00370-t003:** Quotations to illustrate the themes from the responses to: ‘Please tell us more about why you think Sarah’s pain is real or not real.’

Themes	Illustrative Quotes from Participants Who Responded ‘Yes’ (That Sarah’s Pain Is Real) (77%)	Illustrative Quotes from Participants Who Responded ‘No’ or ‘It Depends’ (That Sarah’s Pain Is Not Real) (23%)
The meaning and purpose of pain as criteria for ‘real’ for the teacher.	*“If she is feeling pain, her pain is real. That does not mean it has a physical cause that needs treating but it does not make her pain less real.”* *“Whether it’s emotional or of physical cause- pain is still real.”* *“Even if it’s anxiety it’s real. That pain is Sarah’s body telling her that she’s not coping with something.”* *“If it’s intermittent and related to anxiety then yes it would be real especially to her even if not a physical issue like IBS.”* *“Pain, like behavior, is communication. The pain may not actually be *from* her stomach but it is real and needs to be addressed.”* *“Emotional worries (often caused by friendship issues) can present as tummy upsets. Anxiety can too.”*	*“It could be real and be a stress response to the situation that is causing her anxiety or it could be that she is faking feeling sick to get out of the social situation that is making her feel anxious*.” *“Could be an exit strategy or a way out of the next lesson.”* *“Her pain could be a way of getting out of class for the session. Or Her pain could be genuine and she has something ailing her.”* *“Has Sarah presented before and later the school has known that she uses pain as a cover for something else?”* *“Depending on the actual cause would depend on if the pain is real. If it is bullying then the pain could be a subconscious pain. If it is IBS it could be actual pain.”*
The observed pattern and associated behaviors of the student’s pain.	*“If she’s had it before but presenting again and is distressed I would say her pain is real.”* *“She is very distressed and won’t settle tells me that she may be in real pain.”* *“Crying is difficult to fake…”* *“It has been consistent and she seems distressed…”*	*“Since it is something that has happened on a regular basis and she is often upset after lunch then it’s more likely to be anxiety”* *“Her reaction seems intense…”* *“… she could be exaggerating…”* *“I’d want to know how Sarah responds to different conclusions. Is this always before a certain subject or academic event? Or always in response to a social interaction. Is she satisfied when she is removed from class or school?”*

**Table 4 children-10-00370-t004:** Quotations to illustrate the themes from teacher responses: ‘Do you think students who report pain should not attend class?’

Theme	Quotation from Teacher
The teacher’s role and its boundaries	*“…I am a teacher not a Dr…”* (Answered ‘Yes’) *“If medical attention has not been sought, (the) child should not be in class until all physical causes have been ruled out.”* (Answered ‘Yes’) *“As teachers one of our roles is to support our students to understand what causes them stress or worry.”* (Answered ‘No’) *“Teachers can and should be responsive to the student’s needs making modifications and accommodations as necessary.”* (Answered ‘No’) *“You have to use a bit of teacher judgement to see if it can be worked through or not.”* (Answered ‘I don’t know’)
How pain influences learning and the wider classroom	*“Depending on the severity and genuineness of the response, students aren’t going to learn effectively if they are in pain.”* (Answered ‘Yes’) *“They often interrupt class and can disengage others”* (Answered ‘Yes’) *“The issue needs to be addressed, but avoiding the class/activity doesn’t permanently solve the problem.”* (Answered ‘No’) *(You) “need to ascertain the reason as too much time can be missed due to students using pain as an excuse.”* (Answered ‘No’) *“I believe it should be a case-by-case situation. Some students use an unseen pain (stomach ache, headache) as an excuse to miss out on learning.”* (Answered ‘I don’t know’)

**Table 5 children-10-00370-t005:** Results of univariate regression analyses between higher COPI-Proxy total scores (n = 233) and other variables.

Variable	β Coefficient (95%CI)
Gender (Female)	−1.3 (−3.3 to 0.7)
Teaching experience with 10–12-year-old children	0.0 (−0.1 to 0.1)
Do you have persistent pain? (Yes)	−0.8 (−2.0 to 0.4)
Do you have abdominal persistent pain? (Yes)	−0.7 (−3.2 to 1.9)
Higher COPI score (0–56 points)	**0.5 (0.4 to 0.6)**
Do you think Sarah’s pain is real? (Yes)	**1.4 (0.1–2.8)**
Have you encountered children in your classroom that report recurrent or persistent pain? (Yes)	−0.4 (−1.9 to 1.1)
Do you think students who report pain should not attend class? (No, they should stay in class)	**1.2 (0.0 to 2.4)**

Note: Bold indicates a coefficient where the 95% Confidence Interval does not cross 0.

## Data Availability

The data presented in this study are available upon request from the corresponding author. The data are not publicly available as per ethical approval.

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
