# Peer review of "Do Teachers Question the Reality of Pain in Their Students? A Survey Using the Concept of Pain Inventory-Proxy (COPI-Proxy)"

_children, 2023, doi:10.3390/children10020370_

Round 1

Reviewer 1 Report

Thank you for submitting your paper to the Journal.

This is an interesting topic

Ln 58 you decided to use the COPI for adults even though you note previously that the NPQ and PedNPQ had difficulties. I note also that one of the current authors has developed a COPI – Adult.  Can you expand on this further, did you also find difficulties using a questionnaire designed for one group of individuals in another group? Why was the COPI Adult developed?

Do you think using the COPI adult in this study (if it had been available) would have altered your results?

Ln 83 I can guess what snowballing means but please describe your meaning.

Where is the Pain in Childhood list kept please give a reference.

Lne 127

This reviewer has not had access to your video vignette but it sounds like Sarah has “acute” stomach pain after her lunch break. This is different to chronic pain of 3 months duration?

Presumable teachers would react differently to this kind of pain than to someone with > 3months of pain from widespread childhood arthritis? Maybe you need an explanatory sentence somewhere?

Ln 154 and 155, for non-statisticians please briefly explain oblimin rotation and scree plot.

Table 1

Asking a question like Do you think Sarah’s pain is real? Is a pretty loaded questions. It’s good to see that many responses recognised this! Responses varied but some responses reflected good Pain Science Education already.

The other challenging question is:

Do you think students that report pain should not attend class? Yes No I don’t know.

Why didn’t you ask “Do you think students that report pain should attend class?”

Creating a negative is open to confusion. For example Answer No. Does it mean -  No I do not think that students should not attend class? Or does it mean the negative answer mean Yes Students with pain should attend class?  Please comment.

In any event you have reporting a range of good answers reflecting teachers beliefs.

Ln 201 20% of teachers though she might be faking her pain is interesting

Ln 243 Please explain scree plot elbow criteria.

Discussion

You comment that there is variance between teacher’s beliefs about there own pain and that of their students. Can you comment on why this may be? Is it related to lack of Pain science Education? Could education change these beliefs?

Conclusions

Should you mention the COPI-Adult here? Would it change your conclusions?

Author Response

Response to reviewer 1.

Ln 58 you decided to use the COPI for adults even though you note previously that the NPQ and PedNPQ had difficulties. I note also that one of the current authors has developed a COPI – Adult.  Can you expand on this further, did you also find difficulties using a questionnaire designed for one group of individuals in another group? Why was the COPI Adult developed?

Do you think using the COPI adult in this study (if it had been available) would have altered your results?

Response: Thank you. We did consider using the COPI-adult version for the Proxy – however, we considered the most likely clinical use of the tool would be for dyads involving children – in which case we wanted to ensure we had a validated questionnaire for children that we could directly compare results at an item-level between the parent/adult and the child. In terms of results for this study, we don’t think it would have meaningfully changed the psychometric properties of the COPI-Proxy, or the overall findings about a teacher’s concept of their student’s pain. However, using the COPI-Adult would have meant that item-level comparisons with children would not be possible. We have adjusted the wording in the introduction to reflect your comments and suggestions:

The Concept of Pain Inventory (COPI) is a recently developed tool used to assess a child’s concept of pain which allows clinicians and researchers to individualize education and target pain beliefs [14]. The COPI was recently recommended as a superior alternative to the NPQ in future studies involving PSE in pediatrics [13] because it has been developed for children using simpler language and a 5 point Likert scale to assess the level of agreement rather than True /False which indicates correctness. The COPI has been validated in children aged 8-12 years, with acceptable internal consistency (Cronbach’s α= 0.78) and moderate test-retest reliability (intraclass correlation coefficient (3,1) = 0.55; 95% CI, 0.37-0.68) [14]. We deemed it an appropriate tool to modify as a proxy to assess one’s concept of another person’s pain, such as in a teacher and their student, or a parent and their child, because it would allow for a direct item-level comparison between children and their adult influencers when treating these dyads. We called the modified questionnaire the COPI-Proxy.

Ln 83 I can guess what snowballing means but please describe your meaning.

Where is the Pain in Childhood list kept please give a reference.

Response: Thank you – We have expanded this sentence to address these two comments here:

Recruitment using snowballing [29] was attempted by sharing the flyer via the Pain in Childhood email list (https://listserv.dal.ca/index.cgi?A0=PEDIATRIC-PAIN) and re-questing these colleagues to share the flyer with their friends and colleagues to promote increased visibility.

Lne 127

This reviewer has not had access to your video vignette but it sounds like Sarah has “acute” stomach pain after her lunch break. This is different to chronic pain of 3 months duration? Presumable teachers would react differently to this kind of pain than to someone with > 3months of pain from widespread childhood arthritis? Maybe you need an explanatory sentence somewhere?

Response: Thank you, we have added a more detailed vignette description in the manuscript so this is clearer. The full script from the vignette is in the supplementary file. 

In this part of the survey participants were directed to a 20 second video. The video presents a scenario with a child named Sarah experiencing recurrent stomach pain. She presents on that day with stomach pain after lunch, is crying and re-questing to go home.

Ln 154 and 155, for non-statisticians please briefly explain oblimin rotation and scree plot.

Response: We have added more text in here:

For the psychometric properties of the COPI-Proxy, we first assessed the factor structure. A maximum likelihood factor analysis with oblimin rotation was conducted to extract potential factors.  Oblimin rotation was used to allow for correlation. A scree plot was used to indicate the number of factors that should be generated by the analysis. Factor solutions were forced based on visual inspection of a scree plot (i.e. the elbow on the scree plot indicates the number of factors), theoretical groupings, and how well items loaded onto resulting fac-tors [24].

Table 1

Asking a question like Do you think Sarah’s pain is real? Is a pretty loaded questions. It’s good to see that many responses recognised this! Responses varied but some responses reflected good Pain Science Education already. The other challenging question is:

Do you think students that report pain should not attend class? Yes No I don’t know.

Why didn’t you ask “Do you think students that report pain should attend class?”

Creating a negative is open to confusion. For example Answer No. Does it mean -  No I do not think that students should not attend class? Or does it mean the negative answer mean Yes Students with pain should attend class?  Please comment.

Response: Thank you for raising this issue regarding the mistake in our question. Unfortunately, we became aware of this issue after receiving participant responses and as such, it was too late to make any adjustments to the questionnaire. When we analysed the data, the ‘why’ question that immediately followed this was able to give us more insight into how teachers understood the question. We chose to present the answers/data with the emphasis on descriptive data rather than splitting into yes/no/I don’t know (like we did with the previous question) as we believed this to be a more valid way of presenting responses to a question that could have been misinterpreted by the participants. 

In any event you have reporting a range of good answers reflecting teachers beliefs.

Ln 201 20% of teachers though she might be faking her pain is interesting.

Ln 243 Please explain scree plot elbow criteria.

Response: We have added the brackets to explain this further:

The visual inspection with scree plot elbow criteria (i.e. the point at which the downward curve levels off) suggested 1 factor similar to that of the COPI, however 3 items in the COPI-Proxy did not load well onto this single factor (factor loading <0.32) [30]…

Discussion

You comment that there is variance between teacher’s beliefs about there own pain and that of their students. Can you comment on why this may be? Is it related to lack of Pain science Education? Could education change these beliefs?

Response: Thank you for bringing this to our attention. We have clarified this sentence in paragraph 2 of the discussion to reflect this:

Inconsistency in language may reflect a possible gap in understanding the complexity of pain, which could be contributing to pain-related felt stigma in students who experience chronic pain [6,34], despite the best intentions of teachers to scaffold and support their students. Pain science education may provide teachers with a deeper understanding and consistent language to communicate pain experiences, which in turn may reduce stigma in schools.

Conclusions

Should you mention the COPI-Adult here? Would it change your conclusions?

Response: As per the earlier comment we don’t think the COPI-Adult would change the conclusions. But, after considering your comments, we have more clearly emphasised the clinical reasoning for the COPI-Proxy to align with the child version for direct item-level interpretation:

As social and relational factors are integral to a pain experience, the COPI-Proxy could be useful to direct treatment approaches in dyads such as teacher and student as it allows for direct item-level comparisons with the child’s concept of pain. 

Reviewer 2 Report

Thank you for sharing this article with me. This is an interesting study and should be published. I have several minor comments.

1. The authors seem to firmly believe that teachers can suitably play an important role for pain interaction with students, leading to the conclusion that "perhaps there is also a need for teachers to have a deeper understanding of pain to feel autonomous in their role." Is it possible that schools need to allocate pain experts to address the issue of pain interaction?

2. Please discuss some possible policy interventions based on study findings.

3. Due the nature of data collection, self-selection is likely to be a significant limitation. I'd recommend the authors to briefly discuss this as part of the limitation section.

4. I saw several grammar/style issues throughout the manuscript. Please carefully check them again.

Reviewer 3 Report

An interesting study. I take from this that teacher self-perceptions related to pain can influence how they perceive pain in their students, which is perhaps not surprising. But of course it is important to note that teachers are not physicians and cannot make determinations about cause of pain. This feeds  into course and student management, but I think the point needs to be made that if students report pain, policies need to be in place on what to do.

While the response rate appears decent, we cannot calculate an actual response rate, and we cannot see if this population is similar to the larger population of teachers. As noted by the authors, self-selection is a potential problem as a result.

While you discuss research implications I think you are remiss in not discussing policy implications, both at national level and at the level of an individual school.

Reviewer 4 Report

This study examines teachers' perceptions of their students' distress. The study's goal is to help guide preventative and targeted school-based pain science teaching.

It is an intriguing, creative study that may have practical implications for future research.

1.       What are the children's aches and pains that the instructors alluded to?

2.       Do these teachers have any experience working with special students who have ADHD, dyslexia, or autism?

3.       Do these teachers have kids that are emotionally and cognitively sensitive?

4.       Why was just stomach discomfort mentioned, rather than other forms of pain?

5.       Anxiety was a more significant cause of pain. It appears to have been induced in the survey answer. Otherwise, make it clearer or include it as a research constraint.

6.       What is the connection between food allergies and chronic pain? To be more specific.

7.       What is the explanation for having informed the word “Anxiety” in the research? What prompted such a high response rate in the survey? Do these professors have any fundamental training about pupils' emotional and psychological states?

Author Response

Thank you for your review.

This study examines teachers' perceptions of their students' distress. The study's goal is to help guide preventative and targeted school-based pain science teaching. It is an intriguing, creative study that may have practical implications for future research.

  1. What are the children's aches and pains that the instructors alluded to?

The survey responses from teachers are in response to the video vignette. To clarify this, we have expanded the explanation in the methods.

In this part of the survey participants were directed to a 20 second video. The video presents a scenario with a child named Sarah experiencing recurrent stomach pain. She presents on that day with stomach pain after lunch, is crying and requesting to go home.

  1. Do these teachers have any experience working with special students who have ADHD, dyslexia, or autism?

Our team does agree that experience working with students with these developmental vulnerabilities would likely influence teachers’ perceptions of their student’s pain. Similarly,  we thought that perhaps there were many other experiences that might influence perceptions such as working with chronic health conditions like JIA or children who have experienced trauma.  To account for this diversity in experiences, we recruited a large sample size of teachers.

  1. Do these teachers have kids that are emotionally and cognitively sensitive?

We did not explicitly ask participants if they taught children that are emotionally and cognitively sensitive. Limitations to our survey are described in the discussion:

Parts of our survey were exploratory with the aim of understanding our findings. Therefore, analysis of open responses should be interpreted with some caution as teachers were only given the opportunity to report short answers and answering these questions was not mandatory. Further qualitative research to explore lived experience with teachers and students in the classroom would be beneficial to support these results.  

  1. Why was just stomach discomfort mentioned, rather than other forms of pain?

Our team worked with teachers who teach 10-12 year old children to develop the vignette scenario that described a common presentation of pain in that age group. Our desire here was for a pain complaint that would be familiar for teachers and could be the result of many possible causes, so that we could explore teachers’ perceptions more broadly.

  1. Anxiety was a more significant cause of pain. It appears to have been induced in the survey answer. Otherwise, make it clearer or include it as a research constraint.

Anxiety was one of many possible causes for Sarah’s pain that teachers listed in their open-ended survey responses. This initial question was unprompted by the survey. We have described this in the following paragraph:

The results of exploratory questions that we asked teachers are presented in Tables 2, 3 and 4. Table 2 summarizes the responses to the question: ‘What do you think could be the cause of Sarah’s pain?’. Initially participants provided open-ended responses and results from the content analysis are reported as percentages. Of the 181 participants that completed the question, 75% reported anxiety as one possible cause.

  1. What is the connection between food allergies and chronic pain? To be more specific.

We did not aim to evaluate a possible connection between food or other allergies and chronic pain in this exploratory question. Teachers listed food allergies as one of many possible causes of Sarah’s pain in the open-ended survey responses. This was unprompted by the survey. We have included the following paragraph in the limitations to highlight that this part of the survey was exploratory.

Parts of our survey were exploratory with the aim of understanding our findings. Therefore, analysis of open responses should be interpreted with some caution as teachers were only given the opportunity to report short answers and answering these questions was not mandatory. Further qualitative research to explore lived experience with teachers and students in the classroom would be beneficial to support these results.  

  1. What is the explanation for having informed the word “Anxiety” in the research? What prompted such a high response rate in the survey? Do these professors have any fundamental training about pupils' emotional and psychological states?

We agree and these questions have prompted many future research questions.

  • Regarding a possible explanation for why teachers use the word ‘anxiety’ – In the strengths and limitations we suggest future qualitative research with teachers to give some depth to why teachers answered the way they did:

Further qualitative research to explore lived experience with teachers and students in the classroom would be beneficial to support these results.  

  • We also discuss the possibility that teachers may not have the language to support the body/mind link and hence may use stigmatising language (which may include the use of ‘anxiety’ without proper diagnosis):

Teachers may be negatively influencing pain-related trajectories and outcomes for students. Inconsistency in language could reflect a possible gap in understanding the complexity of pain, and this could be contributing to pain-related felt stigma in students who experience chronic pain [6,34], despite the best intentions of teachers to scaffold and support their students. Pain science education may provide teachers with a deeper understanding and consistent language to communicate pain experiences, which in turn may reduce stigma in schools

  • Regarding the response rate: The survey was international and recruited teachers over 9 months via social media. There is no way to know whether the response rate was high, given the potential for many possible views. We have amended the strengths and limitations to clarify this:

However, as with the nature of online surveys, there is potential for bias due to self-selection, which means that we cannot assume that this population is representative of the larger population of teachers.

  • Regarding “whether the professors have any fundamental training about pupils’ emotional and psychological state”, we have no information. We cannot assess this given the nature of the survey. We have discussed these limitations and highlighted the need for further qualitative research to explore teacher perceptions and experiences.